# Federated Learning from Pre-Trained Models: A Contrastive Learning Approach

**Yue Tan**[1]**, Guodong Long**[1]**, Jie Ma**[1]**, Lu Liu**[2]**, Tianyi Zhou**[3,4]**, Jing Jiang**[1]

[1]Australian Artificial Intelligence Institute, FEIT, University of Technology Sydney
[2]Google Research, [3]University of Washington, [4]University of Maryland
yue.tan@student.uts.edu.au, guodong.long@uts.edu.au
jie.ma-5@student.uts.edu.au, lu.liu.cs@icloud.com
tianyizh@uw.edu, jing.jiang@uts.edu.au

## Abstract

Federated Learning (FL) is a machine learning paradigm that allows decentralized clients to learn collaboratively without sharing their private data. However, excessive computation and communication demands pose challenges to current FL frameworks, especially when training large-scale models. To prevent these issues from hindering the deployment of FL systems, we propose a lightweight framework where clients jointly learn to fuse the representations generated by multiple fixed pre-trained models rather than training a large-scale model from scratch. This leads us to a more practical FL problem by considering how to capture more client-specific and class-relevant information from the pre-trained models and jointly improve each client's ability to exploit those off-the-shelf models. In this work, we design a **Fed**erated **P**rototype-wise **C**ontrastive **L**earning (FedPCL) approach which shares knowledge across clients through their class prototypes and builds client-specific representations in a prototype-wise contrastive manner. Sharing prototypes rather than learnable model parameters allows each client to fuse the representations in a personalized way while keeping the shared knowledge in a compact form for efficient communication. We perform a thorough evaluation of the proposed FedPCL in the lightweight framework, measuring and visualizing its ability to fuse various pre-trained models on popular FL datasets.

## 1 Introduction

Federated learning (FL) is a promising field of machine learning that allows multiple clients to train together without sharing their private data [1]. Vanilla FL aims to train a single global model over all participating clients by periodically synchronizing their model parameters. However, the learned model usually does not perform well on all clients due to the statistical heterogeneity among local datasets [2, 3, 4]. Personalized federated learning (PFL) is proposed to solve this problem by training a personalized model for each client. Recent studies on PFL leverage various techniques to enable more common underlying information shared across different clients [5, 6, 7]. So far, FL and PFL have been widely used in computer vision [8], natural language processing [9], graph data mining [10, 11], and some practical applications, e.g., healthcare [12], finance [13], mobile Internet [14, 15], etc.

However, the models in real-world applications are usually large-scale neural networks which incur *high computation costs* and require *high communication bandwidth* when trained from scratch. This can make it infeasible to train such models in some practical FL scenarios, e.g., low-resource device-based federated learning. To alleviate the above issues, we propose a lightweight FL framework that uses *multiple fixed pre-trained backbones* as the encoder, followed by learnable layers to fuse the representations generated by the backbones for each client. The proposed framework is capable of fusing the representations generated by pre-trained models with various architectures or obtained from vari-

36th Conference on Neural Information Processing Systems (NeurIPS 2022).

ous source data, expanding the scope of federated learning by integrating off-the-shelf foundation models. Also, it makes it possible to utilize large-scale pre-trained models, e.g., VisionTransformer [16] and Swin Transformer [17], in a computation resource-constrained case to enhance the overall performance. Using the pre-trained foundation models as the fixed encoder can efficiently reduce costs because neither complicated backward propagation computation nor large-scale neural network transmission between the server and clients is needed during the training stage. As shown in Table 1, compared to training a ResNet18 [18] from scratch using FedAvg [1], a much smaller number of parameters are communicated per round when learning to fuse the representations generated by fixed pre-trained models. Besides, higher performance can be achieved when using pre-trained models after the same communication rounds. Compared with using a single pre-trained model, using multiple models pre-trained on diverse datasets can provide a more comprehensive view for an input sample [19].

To enable a better personalized representation ability for each client under this lightweight FL framework, we need to select an appropriate information carrier to share common underlying knowledge across clients. Motivated by [22, 23, 24], class-wise prototypes, defined as "a representative embedding" for a specific class, can be an effective information carrier for communication between the server and clients. Sharing prototypes allows for better knowledge sharing across various

Table 1: Compared with training a single ResNet18 from scratch, less training time per round and fewer learnable parameters are needed when using multiple fixed ResNet18 pre-trained on different datasets [20]. Experiments are implemented with FedAvg [1] on Digit-5 dataset [21] under feature shift non-IID setting [3]. The number of communication rounds is 50.

| Mode | Param.↓ | Time↓ | Acc↑ |
|---|---|---|---|
| Training from scratch | 11M | 0.95s | 31.75(3.07) |
| Using 1 pre-trained model | 0.13M | 0.31s | 38.65(2.65) |
| Using 3 pre-trained models | 0.40M | 0.64s | 40.47(3.05) |

ous learning domains, which has been proved in transfer learning [25] and multi-task learning [26] scenarios. To efficiently extract the useful shared information learned from the pre-trained models via prototypes, we design an algorithm called **Fed**erated **P**rototype-wise **C**ontrastive **L**earning (FedPCL) where both local prototypes and global prototypes are used for knowledge sharing in a supervised contrastive manner. By maximizing the agreement between the fused representation and its corresponding prototypes with contrastive learning, class-relevant information and semantically meaningful knowledge are captured by each client. Concretely, global prototypes force the fused representation to be closer to the global class center, while local prototypes force clients to share more higher-level feature information in a pairwise way. Using prototypes to realize inter-client communication allows for knowledge sharing in a latent space and brings additional benefits. Compared with directly transmitting the representation of a sample, prototypes can eliminate bias from a single sample and protect clients' privacy. Compared with model parameters, prototypes are much smaller in size, which significantly reduces communication costs.

We quantitatively evaluate the performance of FedPCL and state-of-the-art FL algorithms under the lightweight framework based on benchmark foundation models and datasets. We also perform extensive experiments to validate the effectiveness of FedPCL in fusing representations output by different backbones and its capability to integrate knowledge from backbones with different model architectures. Our main contributions are summarized as follows:

- We take the first step towards integrating pre-trained models into federated learning to form a lightweight FL framework, which substantially reduces computation and communication costs of current FL frameworks.
- We further propose FedPCL, a novel algorithm that uses class prototypes as the information carrier and conducts contrastive learning during local updates, which allows clients to share more class-relevant knowledge from both local and global prototypes.
- Experiments are conducted on a variety of benchmark foundation models and datasets to measure FedPCL's ability to fuse various pre-trained models for each client. The results indicate that FedPCL outperforms baselines with a better personalization and knowledge integration ability.

## 2 Related Work

**Personalized Federated Learning.** Personalized Federated Learning (PFL) aims to train a personalized model for each client in the FL framework so that the model can achieve better performance on the local dataset. Existing approaches for PFL are based on various techniques. [27, 28, 29] add

an additional term to the original loss function of each client to produce better personalized models according to the private data. [3, 7] share part of the model and keep personalized layers private to achieve personalization. [30] proposes to use a central hypernetwork to generate personalized models for clients. [31] enables a more flexible personalization by adaptively weighted aggregation. [5, 32] study PFL from a meta-learning perspective where a meta-model is learned to generate the initialized local model for each client.

**Contrastive Learning.** Contrastive learning has been widely applied to self-supervised learning scenarios in recent years, achieving state-of-the-art performance in the unsupervised training of deep image models [33, 34] and graph models [35, 36, 37]. A number of works are focused on learning an encoder where the embeddings of the same sample are pulled closer and those of different samples are pushed apart [38, 39, 40, 41]. [42] extends contrastive learning from self-supervised settings to fully supervised settings, enabling us to better exploit label information with contrastive learning. There are also some works incorporate contrastive learning into federated learning to assist local training to achieve higher model performance [43, 44, 45].

**Prototype Learning.** Prototypes have been widely used in a variety of tasks in transfer learning [25], multi-task learning [26], and few-shot learning [22, 46, 47]. It is usually defined as the mean feature vectors of samples within the same class [48, 49]. The authors in [50] represent task-agnostic information by prototypes for distributed machine learning systems and propose a multi-task model fusion method that integrates prototypes for a new task. Since the prototype has the ability to generalize semantic knowledge from similar samples, it is used to assist the local training in federated learning by several studies [51, 24, 44, 43].

**Pre-Trained Foundation Model.** Due to the huge number of parameters and the broad data available for training, pre-trained foundation models can better capture knowledge for downstream tasks and lead to green AI [52, 53]. Recently, pre-trained models (e.g., ViT [16], DETR [54], BERT [55]) have been widely investigated in both vision and natural language processing (NLP) tasks [56]. Extracting task-specific knowledge from the pre-trained models has the potential to achieve the state-of-the-art performance due to their generality and adaptability to different tasks [57, 58, 59]. However, how to integrate the pre-trained models into a federated learning paradigm and keep the whole framework lightweight still remains an open problem.

## 3 Problem Formulation

In this section, we formulate our proposed lightweight FL framework which integrates off-the-shelf pre-trained models as fixed backbones and learns to fuse them adaptively. We start from the general framework of FL, and then explicitly define the problem and explain the proposed global objective.

**General FL Framework.** Formally, the global objective of general FL across $m$ clients is

$$\min_{(w_1, w_2, \cdots, w_m)} \frac{1}{m} \sum_{i=1}^{m} \frac{|D_i|}{N} L_i(w_i; D_i) \tag{1}$$

where $L_i$ and $w_i$ are the local loss function and model parameters for client $i$, respectively. $D_i$ is the private dataset of the $i$-th client. $N$ is the total number of samples among all clients.

The objective of general FL, such as vanilla FedAvg and its variants, is to learn an optimal global model $w$ across $m$ clients, where $w = w_1 = w_2 = \cdots = w_m$ [1]. This can be achieved by periodically synchronizing the model parameters of all clients at the server. However, parameter synchronization sometimes deteriorates the performance on local datasets in the presence of data heterogeneity [7]. Some recent studies [6, 60] also explore personalized FL by applying various constraints and regularization terms, which allows clients to keep different models $w_i, i \in [1, m]$ to achieve higher performance on their local datasets.

**The Proposed Lightweight FL Framework.** Similar to most FL frameworks, there are $m$ clients and a central server involved in the multiple pre-trained backbone-based framework. Each client $i \in [1, m]$ owns $K$ shared and fixed backbones and a private dataset $D_i$ that cannot be shared with each other. Each learning model can be seen as a combination of at least two parts: (i) *Feature Encoder* $r(\cdot; \Phi^*) : \mathbb{R}^d \to \mathbb{R}^{K \times d_e}$, comprising $K$ fixed pre-trained backbones, each of which maps the raw sample $\mathbf{x}$ of size $d$ to a representation vector of size $d_e$. $K$ representation vectors are concatenated together as the output $r(\mathbf{x}; \Phi^*)$, denoted as $r_{\mathbf{x}}$ for short. (ii) *Projection Network* $h(\cdot; \theta_i) : \mathbb{R}^{K \times d_e} \to \mathbb{R}^{d_h}$, which fuses the $K$ representation vectors and maps $r_{\mathbf{x}}$ from one latent space to another for further representation learning. The formal definition is provided as follows.

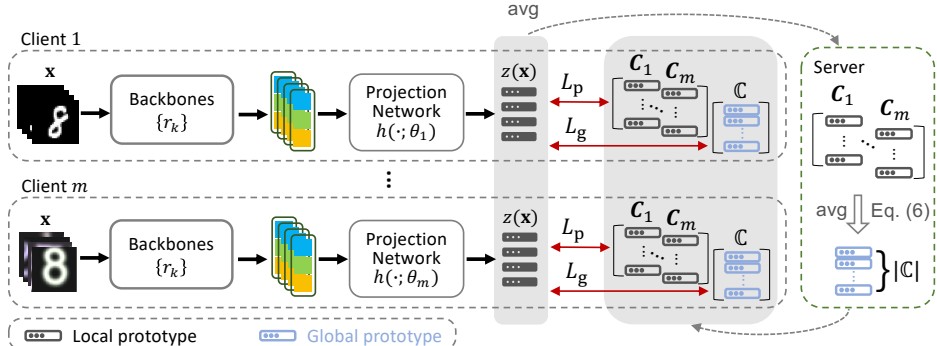

Figure 1: An overview of the proposed lightweight federated learning framework. This example assumes that for each client, there are three pre-trained backbones, with the block in different colors illustrating their backbone-specific representation.

**Definition 3.1.** Let $\phi_k^*$, where $k \in [1, K]$, denote the optimal parameter of the $k$-th backbone pre-trained on a specific dataset, $r_k$ be the embedding function of the $k$-th backbone, and $\mathbf{x}$ denote an instance sampled from a local dataset. We define the *concatenated representation* output by the feature encoder as

$$r(\mathbf{x}; \Phi^*) := \text{concat}\left(r_1(\mathbf{x}; \phi_1^*), \ldots, r_K(\mathbf{x}; \phi_K^*)\right). \tag{2}$$

For client $i$, the projection network $h$, parameterized by $\theta_i$, aims to fuse the representations output by multiple backbones into another abstract space. The output of the projection network is computed as

$$z(\mathbf{x}) = h\left(r_{\mathbf{x}}; \theta_i\right). \tag{3}$$

Based on the above definition, we formulate the global objective of the lightweight FL framework as

$$\min_{\{\theta_1, \theta_2, \ldots, \theta_m\}} \sum_{i=1}^m \frac{|D_i|}{N} \mathbb{E}_{(\mathbf{x}, y) \in D_i} \left[L_i\left(\theta_i; z\left(\mathbf{x}\right), y\right)\right]$$
$$\text{s.t. } z(\mathbf{x}) = h\left(r(\mathbf{x}; \Phi^*); \theta_i\right) \text{ where } \Phi^* = \{\phi_1^*, \phi_2^*, \cdots, \phi_K^*\}. \tag{4}$$

The target of the framework is to learn the personalized projection network $\{\theta_i\}_{i=1}^m$ for each client. Given an input sample $\mathbf{x}$, the representations output by pre-trained backbones are concatenated together as $r(\mathbf{x}; \Phi^*)$. Then, the projection network $h(\theta_i)$ optimized for each client converts the concatenated representation to $z(\mathbf{x})$.

The proposed multi-backbone setting is friendly to most FL methods. The learnable parameter $\theta$ in the projection network can be fully or partly shared across clients for different purposes. For example, FedAvg [1] and its variants [61, 62, 63, 64] can synchronize $\theta$ every round, and personalized FL methods can also adapt to this framework without much modification [27, 7, 65].

## 4 Federated Prototype-wise Contrastive Learning (FedPCL)

In this section, we elaborate our proposed algorithm **Fed**erated **P**rototype-wise **C**ontrastive **L**earning (FedPCL), which is illustrated in Figure 1. In each client, $z(\mathbf{x})$ is generated by the projection network which fuses the representations from multiple backbones. Then, to share the common underlying knowledge across clients, we employ a prototype-based communication scheme to transmit and aggregate prototype sets between the server and clients. With the prototypes returned from the server, we perform local optimization via well-crafted prototype-wise contrastive loss function, which extracts class-relevant information while sharing more inter-client knowledge in the latent space. We provide detailed illustration for each procedure in the rest of this section.

**Prototype as the Information Carrier.** To capture more class-relevant information and semantically meaningful knowledge, we propose to transmit prototypes between the server and clients. Compared with transmitting the learnable model parameters, there are several advantages brought by transmitting class-wise prototypes. Firstly, prototype is more compact in form, which significantly decreases communication costs required during the training process. Secondly, non-parametric communication

allows each client to learn a more customized local model without synchronizing parameters with others. Thirdly, prototypes are high-level statistic information rather than raw features, which raise no additional privacy concerns to the system and are robust to gradient-based attacks [66, 67].

To extract useful class-relevant information from the local side, we construct a local prototype as the information carrier for knowledge uploading. Specifically, it is defined in the latent space of projection network's output by the mean of the fused representations within the same class $j$,

$$C_i^{(j)} := \frac{1}{|D_{i,j}|} \sum_{(\mathbf{x},y) \in D_{i,j}} z(\mathbf{x}), \tag{5}$$

where $D_{i,j}$ refers to the subset of $D_i$ composed of all instances belonging to class $j$, and $\boldsymbol{C}_i$ denotes the local prototype set of the $i$-th client. After the above computation, the local prototype set of each client is sent to the central server for knowledge aggregation, which shares the local class-relevant information extracted on each specific client based on its local dataset.

**Server Aggregation.** After receiving local prototype sets $\{\boldsymbol{C}_i\}_{i=1}^m$ from all participating clients, the server first computes the global prototype as

$$\bar{C}^{(j)} := \frac{1}{|\mathcal{N}_j|} \sum_{i \in \mathcal{N}_j} \frac{|D_{i,j}|}{N_j} C_i^{(j)}, \tag{6}$$

where $\mathcal{N}_j$ denotes the set of clients that own instances of class $j$, and $N_j$ denotes the number of instances belonging to class $j$ over all clients. The global prototype set is denoted as $\mathbb{C} = \{\bar{C}^{(1)}, \bar{C}^{(2)}, \dots\}$. With such an aggregation mechanism, the global prototype set summarizes coarse-grained class-relevant knowledge shared by all clients, which provides a high-level perspective for representation learning.

After aggregation, the server sends both the global prototype set and the full local prototype sets collected from all clients back to every clients. For the situation where only a few classes in each client in some non-IID cases, we introduce a prototype padding procedure in the server to ensure that each local prototype set contains prototypes corresponding to all classes:

$$C_i^{(j)} = \begin{cases} C_i^{(j)}, & i \in \mathcal{N}_j \\ \bar{C}^{(j)}, & i \notin \mathcal{N}_j \end{cases}. \tag{7}$$

Prototype-based communication and aggregation allow each client to own a unique projection network that is able to fuse the general representations in a customized way. The returned local prototype sets can encourage a mutual learning from a client-relevant perspective while the global prototype set, where each element indicates a class center in the overall data, provides a chance to learn from a highly summarized client-irrelevant perspective.

**Local Training.** After receiving the prototype sets from the server, the main target of local training is to efficiently extract useful knowledge from the local and the global prototypes, respectively, so as to maximally benefit local representation learning. To achieve that, we propose a prototype-wise supervised contrastive loss that consists of two terms, i.e., global term and local term.

To force the fused representation $z(\mathbf{x})$ generated by the local projection network to be closer to its corresponding global class center so as to extract more class-relevant but client-irrelevant information, we define the global prototype-based loss term as

$$L_{\mathrm{g}} = \sum_{(\mathbf{x},y) \in D_i} -\log \frac{\exp\left(z_{\mathbf{x}} \cdot \bar{C}^{(y)}/\tau\right)}{\sum_{y_a \in A(y)} \exp\left(z_{\mathbf{x}} \cdot \bar{C}^{(y_a)}/\tau\right)}. \tag{8}$$

where $z_{\mathbf{x}}$ represents $z(\mathbf{x})$ for short, $A(y) := \{y_a \in [1, |\mathbb{C}|] : y_a \neq y\}$ is the set of labels distinct from $y$, $\tau$ is the temperature that can adjust the tolerance for feature difference [68, 41, 42]. For a specific instance $\mathbf{x}$ sampled from $D_i$, we use an inner dot product to measure the similarity between the fused representation $z_{\mathbf{x}}$ and prototypes.

Apart from the global term, to align $z(\mathbf{x})$ with each client's local prototypes by alternate client-wise contrastive learning in the latent space and enable more inter-client knowledge sharing, we define the local prototype-based loss term as

$$L_{\mathrm{p}} = \sum_{(\mathbf{x},y) \in D_i} -\frac{1}{m} \sum_{p=1}^m \log \frac{\exp\left(z_{\mathbf{x}} \cdot C_p^{(y)}/\tau\right)}{\sum_{y_a \in A(y)} \exp\left(z_{\mathbf{x}} \cdot C_p^{(y_a)}/\tau\right)}. \tag{9}$$

Given the ground-truth label $y$ of $\mathbf{x}$, a prototype set can be divided as one positive prototype $C^{(y)}$ and a set of negative prototypes $C^{(y_a)}$. Both $L_g$ and $L_p$ force the representation of a sample to be closer to those positive prototypes and apart from those negative prototypes. $\bar{C}^{(y)}$ in $L_g$ and $C^{(y)}$ in $L_p$ summarize abstract class-relevant information to different granularity, which provides guidance for optimizing the local projection network from different perspectives. For the $i$-th client, the local objective function in Eq. (4) is defined as a combination of $L_g$ and $L_p$ in the following form,

$$L\left(\theta_i; z\left(\mathbf{x}\right), y, \mathbb{C}, \{\boldsymbol{C}_p\}_{p=1}^m\right) = L_g\left(\theta_i; z\left(\mathbf{x}\right), y, \mathbb{C}\right) + L_p(\theta_i; z\left(\mathbf{x}\right), y, \{\boldsymbol{C}_p\}_{p=1}^m). \quad (10)$$

At the end of the local training in each round, clients upload their local prototype set $\boldsymbol{C}_i$ to the server. We present the FedPCL algorithm in detail in Algorithm 1.

**Prototype-based Inference.** After adequate optimization, in each client, the local prototypes generated by projection network not only contain compact local class-wise information but also absorb general knowledge from all participating clients. Considering their powerful representation capability, we leverage the local prototypes to predict unknown labels in the inference stage. Concretely, for a test sample, we first calculate the similarity scores between the output of projection network and every local prototypes. Then, the predicted result is the class with the maximum similarity score.

**Generalization Bound.** We provide insights into the performance analysis of FedPCL. A detailed description and derivations can be found in Appendix C.

---

**Algorithm 1** FedPCL

**Input:** $D_i, \theta_i, i = 1, \cdots, m$, and $K$ pre-trained backbones with parameters $\phi_1^*, \phi_2^*, \cdots, \phi_K^*$, respectively.
**Server executes:**
1: Initialize prototype sets $\{\boldsymbol{C}_p\}_{p=1}^m$.
2: **for** each round $T = 1, 2, ...$ **do**
3:      **for** each client $i$ **in parallel do**
4:          $\boldsymbol{C}_i \leftarrow$ LocalUpdate$(i, \mathbb{C}, \{\boldsymbol{C}_p\}_{p=1}^m)$
5:      **end for**
6:      Update global prototype by Eq. (6).
7: **end for**
**LocalUpdate**$(i, \mathbb{C}, \{\boldsymbol{C}_p\}_{p=1}^m)$:
1: **for** each local epoch **do**
2:      **for** each batch in $D_i$ **do**
3:          Compute $L_g$ by Eq. (8) with global prototypes.
4:          Compute $L_p$ by Eq. (9) with local prototypes.
5:          Update $\theta_i$ by Eq. (10).
6:      **end for**
7: **end for**
8: Compute local prototypes by Eq. (5).
9: **return** $\boldsymbol{C}_i$

---

**Theorem 4.1.** (Generalization Bound of FedPCL.) *Consider an FL system with $m$ clients. Let $\mathcal{D}_1, \mathcal{D}_2, \cdots, \mathcal{D}_m$ be the true data distribution and $\widehat{\mathcal{D}}_1, \widehat{\mathcal{D}}_2, \cdots, \widehat{\mathcal{D}}_m$ be the empirical data distribution. Denote the projection network $h$ as the hypothesis from $\mathcal{H}$ and $d$ be the VC-dimension of $\mathcal{H}$. The total number of samples over all clients is $N$. Then, with probability at least $1 - \delta$:*

$$\max_{(\theta_1, \theta_2, ..., \theta_m)} \left| \sum_{i=1}^m \frac{|D_i|}{N} L_{\mathcal{D}_i}\left(\theta_i; \mathbb{C}, \{\boldsymbol{C}_p\}_{p=1}^m\right) - \sum_{i=1}^m \frac{|D_i|}{N} L_{\widehat{\mathcal{D}}_i}\left(\theta_i; \mathbb{C}, \{\boldsymbol{C}_p\}_{p=1}^m\right) \right|$$
$$\leq \sqrt{\frac{N}{2} \log \frac{(m+1)|\mathbb{C}|}{\delta}} + \sqrt{\frac{d}{N} \log \frac{eN}{d}}. \quad (11)$$

Theorem 4.1 indicates that compared with the model trained on an ideal data distribution, the performance of the empirically trained model is associated with the VC-dimension of $\mathcal{H}$ and the size of the prototype set $\mathbb{C}$. An expected performance gap can be achieved by using appropriate projection network and number of classes. The generalization bound is also important to ensure a satisfying performance of FedPCL, especially when applying it to some safety-critical scenarios [60, 69, 70].

## 5 Experiments

### 5.1 Experimental Setup

**Datasets and Non-IID Settings.** We evaluate our proposed framework on the following three benchmark datasets: Digit-5 [21], Office-10 [71], and DomainNet dataset [72]. **Digit-5** [21] is a benchmark dataset for digit recognition, including five benchmark datasets, namely SVHN, USPS, SynthDigits, MNIST-M, and MNIST. **Office-10** [71] is a standard benchmark dataset consisting of four datasets, namely Amazon, Caltech, DSLR and WebCam. **DomainNet** [73] is a large-scale dataset consisting of six datasets, namely Clipart, Info, Painting, Quickdraw, Real, and Sketch.

Table 2: Test accuracy under the feature shift non-IID setting. In the column labeled BB (short for backbone), *s* is for a single pre-trained backbone and *m* is for multiple pre-trained backbones. # of Comm Params refers to the average number of parameters sent from a client to the server per round.

| BB | Method | MNIST | SVHN | USPS | Synth | MNIST-M | Avg | # of Comm Params |
|----|--------|-------|------|------|-------|---------|-----|------------------|
| *s* | FedAvg | 70.65(1.15) | 17.10(0.20) | 70.24(1.62) | 32.90(0.75) | 29.33(1.18) | 44.04(0.98) | 133,632 |
|    | pFedMe | 71.13(3.63) | 13.18(1.78) | 69.20(0.30) | 36.25(3.35) | 25.25(2.25) | 43.00(2.26) | 133,632 |
|    | PerFedAvg | 52.68(7.03) | 16.28(1.23) | 53.66(6.58) | 29.05(3.45) | 24.38(2.38) | 35.21(4.13) | 133,632 |
|    | FedRep | 64.00(2.20) | 17.88(1.08) | 70.44(1.27) | 36.50(1.55) | 31.90(0.05) | 44.14(2.03) | 131,072 |
|    | FedProto | 80.40(2.75) | 17.03(0.38) | 88.47(0.91) | 40.90(1.10) | 32.85(0.75) | 51.93(1.18) | 2,560 |
|    | Solo | 60.40(2.25) | 15.60(0.20) | 75.28(4.48) | 34.65(0.05) | 28.48(0.53) | 42.88(1.50) | - |
|    | Ours | **82.75**(0.40) | **18.12**(0.42) | **88.82**(0.15) | **41.40**(0.60) | **33.05**(0.95) | **52.83**(0.21) | 2,560 |
| *m* | FedAvg | 71.68(2.93) | 18.45(0.45) | 72.95(0.86) | 37.35(1.35) | 33.70(2.55) | 46.83(1.63) | 395,776 |
|    | pFedMe | 67.45(2.70) | 15.43(0.38) | 65.66(7.20) | 33.55(4.60) | 31.80(0.20) | 42.78(3.01) | 395,776 |
|    | PerFedAvg | 56.03(2.73) | 17.03(0.63) | 57.55(0.27) | 34.90(2.80) | 30.98(1.53) | 39.30(1.59) | 395,776 |
|    | FedRep | 77.25(1.75) | 16.40(0.50) | 80.25(0.32) | 37.63(2.18) | 36.53(0.28) | 49.61(1.05) | 393,216 |
|    | FedProto | 83.78(0.83) | 17.90(0.10) | **91.74**(0.00) | 43.70(2.45) | 36.43(1.58) | 54.71(0.99) | 2,560 |
|    | Solo | 70.43(4.63) | 15.00(0.40) | 84.90(0.24) | 37.18(2.73) | 34.35(2.20) | 48.37(2.04) | - |
|    | Ours | **84.65**(0.15) | **19.38**(0.63) | 90.74(0.53) | **44.73**(0.37) | **37.25**(0.28) | **55.34**(0.34) | 2,560 |

To mimic non-IID scenarios in a more general way, we investigate three different non-IID settings: (i) *Feature shift* non-IID: The datasets owned by clients have the same label distribution but different feature distributions. (ii) *Label shift* non-IID: The datasets owned by clients have the same feature distribution but different label distributions which is simulated by Dirichlet distribution with parameter $\alpha$ [69]. (iii) *Feature & Label shift* non-IID: The datasets owned by clients are different in both label distribution and feature distribution, which is more common but challenging in real-world scenarios. Details about the non-IID settings can be found in Appendix A.1.

**Baselines and Implementation.** We compare our proposed method with popular FL algorithms including FedAvg [1], pFedMe [27], PerFedAvg [5], FedRep [7], FedProto [24], and Solo, i.e., training independently within each client. In feature shift and label shift non-IID setting, the number of clients is 5. In feature & label shift setting, the number of clients is 5, 4, 6 for Digit-5, Office-10, and DomainNet, respectively.

Similar to [19, 20], we use the ResNet18 [18] pre-trained on Quick Draw, Aircraft, and CU-Birds [20] as the backbones. For all baseline methods, the backbone module is followed by two fully connected layers, corresponding to projection network and classifier, respectively, whereas for our method, the backbone module is followed by only one fully connected layer as the projection network. The output dimension of each backbone and the projection network are 512 and 256, respectively. Note that for a fair comparison, all baselines use the same network architecture on top of the frozen backbones as FedPCL except their essential classifier.

We use a batch size of 32, and an Adam [74] optimizer with weight decay 1e-4 and learning rate 0.001. The default setting for local update epochs is $E = 1$ and the temperature $\tau$ is 0.07. We implement all the methods using PyTorch and conduct all experiments on one NVIDIA Tesla V100 GPU. Details about the model and each dataset can be found in Appendix A.1.

## 5.2 Performance Comparison

Table 2 and Table 3 report the results of our method and baselines in mean (std) format over clients with three independent runs. Table 2 shows the results of two cases: (1) *s*: a single backbone is available; (2) *m*: multiple backbones are available.

The results suggest that: (1) compared with single backbone cases, multiple pre-trained backbones lead to higher test accuracy in most cases and about a $1\% - 4\%$ test accuracy improvement for our method; (2) apart from locally training each client (Solo), our method achieves relatively smaller deviation across different runs compared with most baselines, which demonstrates that FedPCL is able to fuse the representations in a more stable way; (3) the number of communicated parameters in prototype-based method is much lower than that of the model parameter-based methods.

Table 3: Test accuracy under the feature & label shift non-IID setting for Office-10, under the label shift non-IID setting for DomainNet.

| Method | Office-10 | Domainnet |
|--------|-----------|-----------|
| FedAvg | 33.84(4.59) | 28.09(2.91) |
| pFedMe | 30.00(1.41) | 32.65(0.72) |
| Per-FedAvg | 26.04(1.46) | 34.64(0.54) |
| FedRep | 37.24(1.54) | 48.82(0.55) |
| FedProto | 34.54(2.65) | 44.48(0.58) |
| Solo | 36.38(0.54) | 46.70(0.75) |
| Ours | **41.40**(1.19) | **52.92**(3.47) |

**Robustness to varying levels of heterogeneity.** In the label shift non-IID setting, we use Dirichlet distribution with parameter $\alpha > 0$ to simulate the heterogeneity level over 10 clients. As $\alpha$ becomes smaller, the label distribution becomes more heterogeneous. We evaluate our proposed FedPCL with FedRep and Solo by varying $\alpha \in \{0.5, 1, 2, 5, 10\}$ on 10 clients. Figure 2(a) suggests that FedPCL achieves the best performance for all values of $\alpha$. Moreover, FedPCL has the smallest deviation across several runs, which implies the stable convergence and robustness of FedPCL.

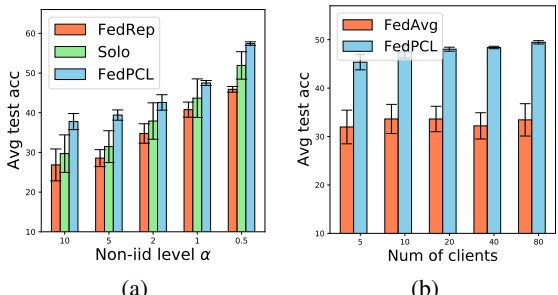

(a)           (b)

Figure 2: Test accuracy on Digit-5 under varying non-IID levels (*left*) and varying numbers of clients (*right*). Both are conducted in the label shift non-IID setting. In (a), non-IID level is controlled by $\alpha$, ranging from 0.5 to 10. In (b), the number of clients ($m$) ranges from 5 to 80.

**Effect of varying numbers of participating clients.** We also compare FedPCL with the vanilla FedAvg by varying the number of clients ($m$) from 5 to 80 as shown in Figure 2(b). The non-IID level remains the same when more clients participate in the training procedure. The results show that the average test accuracy of FedPCL is higher than that of FedAvg by about 15%. Furthermore, the deviation of FedPCL across three runs is much lower than that of FedAvg, especially when $m$ becomes larger. This indicates the scalability of the lightweight framework and the ability of FedPCL to adapt to a large-scale FL system.

## 5.3 Integrating Backbones with Various Architectures

The above experiments use backbones with the same architecture but pre-trained on different source data, showing the capability of FedPCL on integrating knowledge from different source domains. In Table 4, we also show that our proposed lightweight framework is capable of (i) using pre-trained models with different architectures, i.e., a two-layer CNN, AlexNet [75], and VGGNet [76]; (ii) integrating large-scale pre-trained models, i.e., ViT [16, 77], to enhance the performance without the need of huge computation resources to train or fine-tune them. Pre-trained models with different representation abilities should be adaptively leveraged for different tasks to achieve expected performance. They can be selected based on a small set of data before large-scale training.

Table 4: Integrating backbones with various architectures into the proposed framework. Experiments are implemented with FedPCL under feature & label shift non-IID setting.

| Backbone-Domain | Digit-5 | Office-10 |
|---|---|---|
| [ResNet18-QuickDraw, ResNet18-Aircraft, ResNet18-Birds] | 42.87(1.47) | 41.40(1.19) |
| [MLP-ImageNet, AlexNet-ImageNet, VGG11-ImageNet] | **55.80**(2.09) | 70.11(1.78) |
| [tiny-ViT-ImageNet, small-ViT-ImageNet, base-ViT-ImageNet] | 40.96(2.87) | **84.63**(2.57) |

## 5.4 Ablation Study

In this section, we present the results for various ablation experiments to test the effect of global/local prototypes and the contrastive loss. More results can be found in Appendix A.2.3.

**Effect of the contrastive loss.** We test the performance of the local loss function in different forms: (1) Cross Entropy loss; (2) Cross Entropy loss + ProtoDist term, which takes the distance between prototype and fused representation as an additional term to regularize the cross entropy loss; (3) Supervised Contrastive loss that only uses local embedding for contrastive learning; (4) Our prototype-wise contrastive loss that uses both global and local prototypes for local contrastive learning. As shown in Table 5, our prototype-wise supervised contrastive loss outperforms others.

Table 5: Comparison between the cases when different local losses are used for local training. Experiments are conducted on Digit-5 dataset under the feature & label shift non-IID setting where the Dirichlet parameter $\alpha$ is 1, the number of clients is 5, and the number of pre-trained backbones is 3.

| Loss Type | Acc |
|---|---|
| Cross Entropy | 32.98(3.44) |
| Cross Entropy + ProtoDist [24] | 43.94(3.02) |
| Supervised Contrastive [42] | 42.18(3.25) |
| Ours | **45.22(3.65)** |

**Effect of prototypes.** To verify the effectiveness of different kinds of prototypes used for local training in our proposed FedPCL, we compare the following three cases: (i) Only global prototypes computed at the server are used for local training; (ii) Only local prototypes aggregated at the server are used for local training; (iii) Both global and local prototypes are used for local training. All these three cases are implemented under three non-IID settings. As shown in Table 6, without global or local prototypes being used for local supervised contrastive learning, the performance drops 0.3%-2%, indicating that the knowledge conveyed by global prototypes and local prototypes can benefit the local learning framework from different perspectives.

Table 6: Comparison between the cases when only global prototypes are used for local training, only local prototypes from all clients are used for local training, and both of them are used for local training. Experiments are conducted on Digit-5 dataset under three non-IID settings where the Dirichlet parameter $\alpha$ is 1, the number of clients is 5, and the number of pre-trained backbones is 3.

| Prototypes | Feature shift non-IID | Label shift non-IID | Feature & Label shift non-IID |
|---|---|---|---|
| Global only | 54.69(0.14) | 44.75(1.77) | 43.79(4.09) |
| Local only | 55.01(0.10) | 44.52(1.73) | 43.08(3.87) |
| Global and local | **55.34**(0.34) | **45.35**(1.58) | **45.22**(3.65) |

## 5.5 Visualizing the Fusing Results of FedPCL

To better understand how FedPCL fuses the representation output by backbones, we visualize the normalized similarity scores under the feature shift non-IID setting. The heatmap in Figure 3 summarizes the values of the normalized similarity scores, computed by the inner product between local prototypes in the single backbone case and the multi-backbone case. For example, in Figure 3(a), the element $0.5$ on row 1 and column 2 represents the similarity score

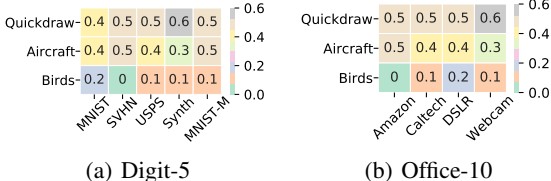

(a) Digit-5        (b) Office-10

Figure 3: Similarity scores generated on two datasets: (a) Digit-5 and (b) Office-10. Rows correspond to the pre-trained backbones $r_k(\mathbf{x})$ and columns correspond to different local datasets with feature shift.

between the local prototype computed when only one backbone pre-trained in Quickdraw is available and the local prototype computed when three backbones pre-trained in Quickdraw, Aircraft, and Birds are available.

We find that the backbone pre-trained on Quickdraw contributes more to the fused prototype, while the backbone pre-trained on Birds contributes the least. Clients who own a specific dataset show different levels of utilization when fusing the representations from various backbones.

## 5.6 Incorporating with Privacy-Preserving Techniques

To further eliminate concerns about privacy leakage, we also incorporate FedPCL with privacy-preserving techniques to observe the variation in performance. Concretely, we add various random noise to the prototypes to be communicated and the original images, respectively. The results show that the performance of FedPCL remains high after noise injection. Details are given in Appendix A.2.4.

## 6 Conclusion

To address the problems on excessive computation and communication demands in current federated learning frameworks, we propose a lightweight framework that leverages multiple neural networks as fixed pre-trained backbones to replace the learnable feature extractor. To customize the general representations generated by these backbones for each client, class-wise prototypes are shared across the clients and the server. To efficiently extract the shared knowledge from the prototypes, we develop FedPCL algorithm that uses contrastive learning at the client-side during the local update. Extensive experiments are conducted to show the superiority of FedPCL under the proposed lightweight framework.

This research has great potential to bring existing FL frameworks to a new paradigm and provide more options on local neural network architectures. However, we mainly focus on the vision task in this work. Methods for language tasks can be further explored in future studies. Considering the evident motivation and effectiveness of the lightweight framework and FedPCL, we believe this work is intriguing for future studies.

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
