# OpenReview forum: "Federated Learning from Pre-Trained Models: A Contrastive Learning Approach"
_NeurIPS.cc/2022/Conference — NeurIPS 2022 Accept_

### Official Review · Reviewer_qfrZ · 2022-06-25

**Rating:** 4
**Confidence:** 4
**Soundness:** 3 good
**Presentation:** 2 fair
**Contribution:** 2 fair

**Summary:**

This paper presents Federated Prototype-wise Contrastive Learning (FedPCL) which aim to exploit pre-trained models in FL environment for reducing communication (COM) costs.


The authors claim the following contributions:
-  First to integrate pre-trained models into federated learning to reduce COM cost.
- Proposing FedPCL.
- Experiments showcasing FedPCL advantages over existing P/FL works.

**Questions:**

See above.

**Ethics Review Area:**

["Privacy and Security (e.g., consent)"]

**Limitations:**

See above.

**Strengths And Weaknesses:**

***Pros:***
- The paper is well written and easy to follow.
- The motivation of this paper is well explained - reducing COM liabilities is a major challenge in P/FL.
- The authors provided generalization bound for FedPCL


***Cons:***
 - The proposed method increases the computational demand from the clients (which in most cases are edge devices e.g. wearables, smartphones, etc.).
- From eq (6,7) I understand that the server will be able to "know" which classes each client has. This raises some privacy issues. For example in medical applications we don't want the server to know which client has/not illness.
- Missing citations of important and strong PFL papers (e.g. [1,2]).
- Problematic results in the experimental part - in lots of cases Solo (local training) outperforms most PFL methods.
- It is not clear if and how the authors performed hyperparameter tuning for FedPCL / baseline approaches. In addition, some experimental details are missing such train/val/test split, etc.
- Experiments were conducted on limited numbers of clients (6 max). Rea-life P/FL systems obtain thousands / millions of clients.


[1] Personalized federated learning using hypernetworks.

[2] Personalized Federated Learning with First Order Model Optimization.

---

> ### Author Response · Authors · 2022-08-02
> **Thank you for your valuable comments! Reply to Reviewer qfrZ**
>
> **Increasing computational demand.**
> Thanks for the comments. Having multiple fixed backbones only increases forward pass computational demand for inference but highly decreases the backpropagation cost during local training since it only needs to train a lightweight model to fuse the pre-trained features. In our proposed framework, feature encoders are frozen during training, so the number of learnable parameters is much smaller than that when training the whole model from scratch, which means less computation demand is required by the backpropagation process. This is a significant advantage since the client usually has a computation bottleneck to afford the local training of large-scale models. To empirically demonstrate this advantage, we report the training time per round in Table 1. It is worthwhile to notice that leveraging pre-trained models actually requires fewer computational resources but achieves higher accuracy after the same number of communication rounds.
>
> **Label-based privacy concern.**
> Thanks for the comments. Label privacy is an open challenge for many existing FL approaches. For example, it is easy to infer the label distribution from the classifier weight, which makes most FL methods face the same issue (Luo et al. 2021). We will study it in the future extensions of our method, e.g., to develop some hashing method to anonymize the labels. In the current method, the label information is shared across clients during the prototype aggregation process in Eq. (6,7). Fortunately, the information conveyed by the label (e.g., image class, illness or not) is usually less sensitive and private than that conveyed by the raw features (e.g., image itself, patient’s detailed data).
>
> Luo et al. No fear of heterogeneity: Classifier calibration for federated learning with non-iid data. NeurIPS 2021
>
> **Missing citations.**
> Thanks for the comments and the references. We have cited them in the Related Work section appropriately.
>
> **Poor performance of PFL methods.**
> Thanks for the comments. It is true that some of the PFL baselines fail to outperform Solo. There are two main reasons for that.
>
> (1) Most of these baselines are designed for the training-from-scratch framework where there are a large number of learnable parameters. The proposed new setting fixes most parameters as the pre-trained models and raises new challenges to existing PFL methods.
>
> (2) When there are feature/label shifts or non-IID distributions across clients, the local performance might be further deteriorated after parameter aggregation. The deterioration can be more significant when only a small number of parameters are locally trained. Since most PFL methods are still based on the parameter aggregation scheme, their performances are inevitably affected.
>
> Due to the page limit, we add the analysis of why some PFL methods do not work well to Appendix B.2.
>
> **Experimental details.**
> Thanks for the comments. We will make these details clear. First, we want to clarify that our work follows the current literature of PFL where clients have different data distributions and train/val/test split is conducted on each client.
>
> Data splitting: We use a portion of data as training samples (~10%) and the rest as test samples. We first take out a 20% subset of the training set for validation to tune the hyperparameters and then use the whole training set to retrain the model (Luo et al. 2021).
>
> Hyperparameter tuning: We use grid search to find the optimal hyperparameters including learning rate, weight decay, and the output dimension of each backbone.
>
> We have clarified this and added the detailed grid search space to Appendix A.1.4 in the revised version.
>
> **Limited numbers of clients.**
> Thanks for the comments. Since we take the first attempt to integrate pre-trained models into FL, most experiments are done on limited numbers of clients. But we have validated the scalability of FedPCL by increasing the number of clients from 5 to 80 as reported in Figure 2(b).

---

> > ### Comment · Reviewer_qfrZ · 2022-08-05
> > **Post Rebuttal**
> >
> > Re increasing computational demand: performing multiple forward passes in P/FL setups may case asynchronous problems which we wish to avoid (as much as possible in FL environments).
> >
> > Re privacy: If lots of people does the same mistake it does not make it right. Specifically in FL we want to avoid privacy issues, therefore mitigation of label usage is high priority. There are lots of works in P/FL which does not use label sharing with the hub.
> >
> > Re limited number of clients: 80 is not enough, most FL systems have thousands (and more) clients.
> >
> > After reading the authors comments I will keep my score.

---

> > > ### Author Response · Authors · 2022-08-08
> > > **Reply to Reviewer qfrZ**
> > >
> > > **Increasing computational demand**: Thanks for the reply. It should be clarified that the multiple forward passes are simultaneously performed and they are fixed at the local side, which means that they are not synchronized at the central server at all. Therefore, there will be no asynchronous concerns.
> > >
> > > **Label privacy**: Thanks for the reply. We are not following the “mistake”. We just want to clarify that, in the FL community, it still remains an open problem whether the label information is privacy or not. Just like we still allow parameter-based transmission even if there have already been studies proving that parameters/gradients can be private information and can be used to reconstruct users’ raw data (Zhu et al. 2019). In our work, we just want to provide a potential communication scheme that uses prototypes for knowledge sharing across clients.
> > >
> > > Zhu, Ligeng, Zhijian Liu, and Song Han. Deep leakage from gradients. NeurIPS 2019
> > >
> > > **Limited number of clients**: Thanks for the reply. Cross-device FL usually refers to the case where there are more than 100 clients and the most famous FL algorithm FedAvg uses 100 clients in their experiments (McMahan, et al., 2017). The number of clients in our experiments is close to 100. We would like to carry out experiments on thousands or even millions of clients, but unfortunately, our computation resources do not allow us to do the real-world large-scale experiments like what google and other companies do in industry.
> > >
> > > McMahan, et al. Communication-efficient learning of deep networks from decentralized data. AISTATS 2017

---

> > > > ### Comment · Reviewer_qfrZ · 2022-08-09
> > > > **post rebuttal**
> > > >
> > > > After reading the authors response I will keep my score unchanged.

---

### Official Review · Reviewer_9cCn · 2022-07-07

**Rating:** 8
**Confidence:** 5
**Soundness:** 4 excellent
**Presentation:** 3 good
**Contribution:** 4 excellent

**Summary:**

This paper proposes a lightweight framework that enables federated learning from multiple pre-trained models, which significantly reduces communication and computation costs and makes it possible to integrate large-scale models into federated learning framework. To efficiently learn and share learned knowledge across clients under this framework, a prototype-based contrastive learning algorithm is developed to fully leverage the representation ability of pre-trained models in the latent space. A comprehensive empirical study is conducted and shows promising results achieved by the proposed algorithm.


**Questions:**

1. Existing SOTA personalized FL methods that achieve high performance in traditional non-IID settings, e.g., pFedMe, FedRep, do not work well under this lightweight framework. What is their limitation when using pre-trained models? Can this be fixed by introducing some simple but efficient techniques?

2. Is it possible to allow clients to have different pre-trained models and keep sharing knowledge in this manner? This would be more common in real-world cases.


**Limitations:**

The authors have well addressed the limitations of this work in terms of specific target tasks and illustrated the potential directions of their future studies.

**Strengths And Weaknesses:**

Pros:
1. The problem studied in this paper is interesting and important. This work first proposes to integrate pre-trained models into federated learning to form a lightweight FL framework, which expands the scope of current FL frameworks and makes an important contribution to the FL community.

2. This paper has its originality and technically sound. The main claims regarding the proposed setting are well supported in the methodology and experimental parts. By transmitting prototype sets and optimizing the well-designed prototype-wise contrastive loss, class-specific knowledge can be fully extracted and shared during local training and server aggregation procedures, respectively.

3. Extensive and comprehensive experiments are conducted. Experimental results on three benchmark datasets under different non-IID settings show superior or comparable performance achieved by FedPCL compared to existing FL/PFL methods. Sufficient ablation studies are provided to demonstrate the advantages of the loss and prototypes in the approach. Detailed data and model settings are provided in the appendix for reproduction.


Cons:
1. It is better to provide more analysis on the performance improvement achieved by FedPCL in Section 5.2 and 5.4 to let the readers well understand how contrastive loss and prototypes make their contribution to the improvement.

2. Although there is a relatively thorough literature review in the related work part, I prefer to see a discussion on the relation of this work and the fine-tuning techniques in FL since both of them have the potential to integrate pre-trained foundation models into the framework.

---

> ### Author Response · Authors · 2022-08-01
> **Thank you for your valuable comments! Reply to Reviewer 9cCn**
>
> We thank the reviewer for the positive review and constructive comments. We provide our responses as follows.
>
> **More analysis on the improvement made by contrastive loss and prototypes.**
> Thanks for the comments. To better explain the contribution of the contrastive loss and the prototypes, we add the following discussion to Section 4 and Appendix B.2 in the revised version.
>
> “Both $L_{\rm g}$ and $L_{\rm p}$ force the representation of a sample to be closer to those positive prototypes and apart from those negative prototypes. $\bar{C}^{(y)}$ in $L_{\rm g}$ and ${C}^{(y)}$ in $L_{\rm p}$ summarize abstract class-relevant information to different granularity, which provides guidance for optimizing the local projection network from different perspectives.”
>
> “Instead of synchronizing learnable parameters, our method allows each client to keep their own local parameters while extracting shared knowledge only by contrastive learning. Prototype is used as the information carrier to achieve that.”
>
> **Relation to the fine-tuning techniques in FL.**
> Thanks for the comments. There is indeed a branch of FL studying fine-tuning techniques for FL (Cheng et al. 2021, Zhang et al. 2022). These works mainly focus on how to adjust the local/global model to improve its representation ability on biased/generic data distribution, while our work utilizes fixed pre-trained foundation models and focuses on improving the fusing ability. Due to the page limit, we will add more discussion on this point to Appendix B.2.
>
> Cheng et al. Federated Asymptotics: a model to compare federated learning algorithms. arXiv preprint 2021
>
> Zhang et al. Fine-tuning global model via data-free knowledge distillation for non-iid federated learning. CVPR 2022
>
> **Limitations of SOTA personalized FL methods.**
> Thanks for the comments. It is true that some SOTA PFL methods do not work well under our proposed lightweight framework. Their performance is mainly due to the following two reasons.
> (1) Most of these baselines are designed based on the training-from-scratch framework where there are a large number of learnable parameters. The proposed new setting where most parameters are fixed is not friendly to some PFL methods.
> (2) When there are feature/label shift non-IID across clients, the local performance might be further deteriorated after parameter aggregation. The deterioration can be more significant when only a small number of parameters are locally trained. Since most PFL methods are still based on the parameter aggregation scheme, their performances are inevitably affected.
>
> Performance might be enhanced by decreasing the number of shared parameters, optimizing local training schemes, and introducing fine-tuning procedures during local training, etc.
>
> **Is it possible to use different pre-trained models?**
> Thanks for the comments. Having different pre-trained models or multiple random initialized feature encoders across clients sounds interesting and promising, we will consider it in our future work. We have added a discussion on applying FedPCL to a wider range of scenarios in Appendix B.1.

---

### Official Review · Reviewer_TqKW · 2022-07-12

**Rating:** 5
**Confidence:** 3
**Soundness:** 3 good
**Presentation:** 3 good
**Contribution:** 3 good

**Summary:**

This paper studies a specific federated learning problem, which considers to leverage the large-scale pre-trained models. It builds a federated prototype-wise contrastive learning framework to capture more client-speciﬁc and class-relevant information. A generalization analysis has been provided to discuss the effect of the built prototypes. The authors conduct a range of experiments to demonstrate the performance of the proposed method over state-of-the-art performance.

**Questions:**

Please explain the questions in the weakness part listed above.

**Ethics Review Area:**

["I don’t know"]

**Strengths And Weaknesses:**

Strengths
(1) By constructing a FedPCL mechanism, the proposed framework trains the model in a lightweight manner and simultaneously shares the knowledge across the clients. Besides, the projection network provides the personalized feature selection for each client.

(2) A theoretical analysis points out the how the proposed prototype-based learning method affects the generalization bound, which helps how this type of methods differ from previous non-prototype-based works.

(3) A range of experiments have been conducted to show the promise of the proposed method over the state-of-the-art methods, and FedPCL can be further boosted by integrating different pretrained models.

Weakness
(1) The relationship between two loss functions in Eq.(8) and Eq.(9) have not sufficiently explained, since the latter provide the whole supervision information about the global C. In this perspective, why we need Eq.(8).

(2) Another question is about the pre-trained backbones. It seems that the proposed framework actually can be applied to the wider scenarios where there are not pre-trained models as the feature extractors. Limiting the model to this scenario raises the concerns that whether the prototype-based federated learning methods easily overfits the local data without the auxiliary feature extractors. The authors could discuss more about this point.

(3) It lacks of a baseline like [1] and the authors should discuss how your method is different from this work and other prototype-based methods like FedProto and why yours is better.

[1] FedProc: Prototypical Contrastive Federated Learning on Non-IID data

---

> ### Author Response · Authors · 2022-08-02
> **Thank you for your valuable comments! Reply to Reviewer TqKW**
>
> **Relationship between two loss functions.**
> Eq. (8) and Eq. (9) utilize global and local prototypes to align the local fused representations, respectively. Although the global prototypes can be obtained by linearly aggregating local prototypes, the following contrastive loss computation is non-linear. Therefore, Eq. (9) alone cannot provide full supervision information during local training. We have empirically studied the effect of these two losses in Section 5.4 and the results in Table 6 show that both the prototypes/losses have critical contributions to performance improvement.
>
> We add the following discussion to Section 4 in the revised version to better explain the relationship between the two loss functions.
>
> “Both $L_{\rm g}$ and $L_{\rm p}$ force the representation of a sample to be closer to those positive prototypes and apart from those negative prototypes. $\bar{C}^{(y)}$ in $L_{\rm g}$ and ${C}^{(y)}$ in $L_{\rm p}$ encode abstract class-level information at different levels of granularity, which provide strong guidance at multiple levels for optimizing the local projection network.”
>
> **Narrow application scenarios and overfitting problems.**
> Our proposed framework does require pre-trained models but they are becoming widely available in various areas. This is mainly due to the fact that most types of data, e.g., images, texts, graphs, have corresponding pre-trained models nowadays. For the cases without pre-trained models, using multiple fixed encoders can be an alternative solution which is another interesting problem and can be explored in the future.
>
> We do think incorporating pre-trained models into existing learning frameworks is a promising trend in deep learning, especially when models are becoming larger and larger in scale and hard to train from scratch. So far, the idea to utilize pre-trained foundation models has been proposed for CV and NLP tasks and achieved certain improvements (Han et al. 2021, Chen et al. 2021, You et al. 2021).
>
> As for the overfitting problem, it truly deteriorates the performance of approaches based on the parameter averaging scheme when the amount of learnable parameters is small. That is the principal reason why some baseline methods fail to perform well in the experiments. However, our proposed method alleviates this problem by (1) aggregating prototypes rather than parameters; (2) extracting globally shared knowledge in a contrastive manner rather than synchronizing the model parameter with the central server. The former prevents local performance deterioration while the latter potentially prevents overfitting by softly adjusting the local projection layers. Due to the page limit, we add more detailed discussions to Appendix B.1 in the revised version to make this point clearer.
>
> Han et al. Pre-trained models: Past, present, and future. AI Open 2021
>
> Chen et al. Pre-trained image processing transformer. CVPR 2021
>
> You et al. Logme: Practical assessment of pre-trained models for transfer learning. International Conference on Machine Learning. ICML 2021
>
> **Comparison with other prototype-based methods.**
> Due to the page limit, we add the following analysis to Appendix B.2 to explain the main differences between our method and other prototype-based FL methods, e.g., FedProc and FedProto, and the reason why ours is better in the proposed setting.
>
> “Although prototypical learning and contrastive learning exist in prior work, they are still based on the learnable parameter aggregation scheme (Li et al. 2021, Mu et al. 2021, Michieli et al. 2021) or just use prototypes/contrastive learning to regularize the original local training (Tan et al. 2022). Instead of aggregating and synchronizing learnable parameters, our method allows each client to keep their own local parameters while sharing knowledge only by contrastive learning defined on prototypes. The prototypes are used as the information carrier to achieve that.  ”
>
> “Some state-of-the-art PFL methods fail to perform well mainly due to the following two reasons.
> (1) Most of these baselines are designed for training-from-scratch where there are a large number of learnable parameters. The proposed new setting fixes most parameters as the pre-trained models and raises new challenges to existing PFL methods.
> (2) When there are feature/label shifts or non-IID distributions across clients, the local performance might be worse after parameter aggregation. The deterioration can be more significant when only a few parameters are locally trained. Since most PFL methods are still based on the parameter aggregation scheme, their performances are inevitably affected.”
>
> Li et al. Model-contrastive federated learning. CVPR 2021
>
> Tan et al. FedProto: Federated prototype learning across heterogeneous clients. AAAI 2022
>
> Mu et al. FedProc: Prototypical Contrastive Federated Learning on Non-IID data. arXiv preprint 2021
>
> Michieli et al. Prototype guided federated learning of visual feature representations. arXiv preprint 2021

---

> > ### Comment · Reviewer_TqKW · 2022-08-05
> > **After rebuttal**
> >
> > After the rebuttal, most of my concerns have been addressed. I will tend to accept this work.

---

### Official Review · Reviewer_R1m9 · 2022-07-12

**Rating:** 6
**Confidence:** 4
**Soundness:** 3 good
**Presentation:** 3 good
**Contribution:** 3 good

**Summary:**

This paper proposes a lightweight federated learning framework where clients jointly learn to fuse the representations generated by multiple fixed pre-trained models rather than training a large-scale model.  The paper is generally well-written and in a good shape. Extensive experiments were conducted to show the effectiveness of the proposed FedPCL.



**Questions:**

1. The generalization bound of FedPCL given in Theorem 4.1 is unclear without sufficient analysis. The authors claimed that ”the bound is associated with the complexity of training models and the size of prototype size”. The reviewer believe that this is a quite straightforward observation even without Theorem 4.1. Then, what is the purpose of Theorem 4.1. The reviewer are encouraged to analyze the convergence of FedPCL with the fixed pre-trained models.

2. In addition, the derivations of Theorem 4.1 in the appendix is also not clear. For example, the derivation of (13) by McDiarmid’s inequality and the derivation of (14) by Rademacher complexity are not clear.


**Limitations:**

This work adequately addressed the limitations. The authors developed a lightweight federated learning framework to reduce the computation and communication costs and integrated pre-trained models to extract prototypes for federated aggregation. The is a new try for federated learning.

**Strengths And Weaknesses:**

Strengths: Using multiple pre-trained models to capture client-specific and class-relevant information is new in federated learning. FedPCL enjoys light computation and communication costs, compared to most other federated learning.

Weaknesses:
1.	Apart from the multiple pre-trained models, FedPCL is built on the idea of prototypical learning and contrastive learning, which are not new in federated learning.
2.	The performance of FedPCL heavily relies on the selection of different pre-trained models, limiting its applications to more wide areas. As shown in Table 4, the model accuracy is quite sensitive to the pre-trained models.

---

> ### Author Response · Authors · 2022-08-02
> **Thank you for your valuable comments! Reply to Reviewer R1m9**
>
> **What is the unique novelty of FedPCL?** Thanks for the comments. This paper has provided novel contributions in terms of (1) integrating pre-trained models into federated learning (as the reviewer pointed out) (2) proposing a novel algorithm FedPCL which allows clients to share knowledge via prototype-based local contrastive learning. Although prototypical learning and contrastive learning exist in prior work, they are still based on the learnable parameter aggregation scheme (Li et al. 2021, Mu et al. 2021, Michieli et al. 2021) or just use prototypes/contrastive learning to regularize the original local training (Tan et al. 2022, Li et al. 2021). In contrast, our method first enables clients in FL to extract shared knowledge only by contrastive learning and the prototype is used as the information carrier to achieve that. Due to the page limit, we have clarified this in Appendix B.2.
>
> Li et al. Model-contrastive federated learning. CVPR 2021
>
> Tan et al. FedProto: Federated prototype learning across heterogeneous clients. AAAI 2022
>
> Mu et al. FedProc: Prototypical Contrastive Federated Learning on Non-IID data. arXiv preprint 2021
>
> Michieli et al. Prototype guided federated learning of visual feature representations. arXiv preprint 2021
>
> **Sensitivity to different pre-trained models.** Thanks for the comments. The sensitive performance is due to various representation abilities of pre-trained models, which is consistent with the fact that there is usually a model selection procedure before large-scale training. Usually, better representation ability means higher computation costs. For real-world cases, there will be a tradeoff between performance and efficiency. The “sensitive” performance can be leveraged to guide the selection of pre-trained models. What we want to show in Table 4 is that our framework has the potential to (1) use pre-trained models with different architectures; (2) integrate large-scale pre-trained models to enhance the performance but do not need to spend resources training them. We will make this clearer in Section 5.3.
>
> **Questions about generalization bound and convergence analysis.** Thanks for the comments. We understand the observation seems intuitive and straightforward. However, deriving such a generalization bound (Theorem 4.1) is important to ensure a satisfying performance of the proposed method, especially when applying it to some safety-critical applications (Lin et al. 2020, Zhang et al.2021, Mansour et al. 2020).
>
> As for the convergence analysis, since there are sufficient convergence analyses for the prototype-based communication scheme in a previous work  (Tan et al. 2022), the convergence of FedPCL can be guaranteed following the same theoretical framework without much modification. Besides, most parameters in the local model are from the feature encoder component and are fixed during the training procedure. Only a small number of parameters in the projection network need to be updated. Considering the number of learnable parameters is small, the convergence rate is not a big concern for FedPCL.
>
> As for the derivations of Theorem 4.1, we apologize for not making it clear. In the original version, the derivation of (13) follows from a variant of McDiarmid’s inequality, while the derivation of (14) mainly follows from (a) the definition of Rademacher complexity and (b) Jensen’s inequality.
>
> We add more details to the “Generalization Bound” part in the main body and appendix to (1) illustrate the purpose to derive the bound, (2) provide more details to make the derivation more explicit and understandable.
>
> Tan et al. FedProto: Federated prototype learning across heterogeneous clients. AAAI 2022
>
> Lin et al. Ensemble distillation for robust model fusion in federated learning. NeurIPS 2020
>
> Zhang et al. Parameterized knowledge transfer for personalized federated learning. NeurIPS 2021
>
> Mansour et al. Three approaches for personalization with applications to federated learning. arXiv preprint 2020

---

> > ### Comment · Reviewer_R1m9 · 2022-08-09
> > **After Rebuttal**
> >
> > The author addressed most of my concerns. Thus, I tend to raise my score.

---

### Author Response · Authors · 2022-08-02
**General response**

We sincerely appreciate all reviewers’ constructive comments on our work. We are glad that the reviewers recognized our contributions in terms of a lightweight FL framework integrating pre-trained models (Reviewers R1m9, TqKW, 9cCn), a novel and efficient algorithm FedPCL (Reviewers R1m9, TqKW, 9cCn), and theoretical and empirical analysis on the proposed algorithm (Reviewer TqKW, 9cCn, qfrZ). Both Reviewer R1m9 and qfrZ think the paper is well-written and easy to follow. Both Reviewer TqKW and 9cCn think the experimental part is comprehensive and extensive.

We have carefully modified the paper according to the reviewers’ comments and highlighted the main modifications in the updated version. In the following, we respond to each reviewer in turn. We will incorporate all the feedback in the final version.

---

> ### Author Response · Authors · 2022-08-08
> **To all reviewers: please let us know if you have any further questions**
>
> Dear reviewers and AC,
>
> Thank you for reading our rebuttal. We have tried to address most if not all concerns raised by the reviewers. Please let us know if you have any further questions about our paper or the rebuttal.
>
> Thank you.
>
> Best regards,
> Authors

---

> ### Author Response · Authors · 2022-08-09
> **To all reviewers: please let us know if you have any further questions**
>
> Dear reviewers,
>
> Thank you again for reading our rebuttal. We have tried our best to address most if not all concerns raised by you. Please let us know if you have any further questions about our paper or rebuttal.
>
>
> If our responses have sufficiently addressed your concerns or answered your questions, it will be great if you are willing to kindly reconsider your score.
>
> Thank you.
>
>
> Best regards,
>
> Authors

---

### Meta-Review · Area_Chair_m1s9 · 2022-08-20

**Recommendation:** Accept
**Confidence:** Less certain

**Metareview:**

To train large-scale models for federated learning settings, this paper presents a Federated Prototype-wise Contrastive Learning (FedPCL) approach. FedPCL aims to solve federated learning problems utilizing the representations of multiple fixed pre-trained models. Specifically, it computes the class prototypes of each client, and their average. Then it sends them to each client, and builds contrastive loss function based on them. Experiments show effectiveness of FedPCL under the proposed lightweight framework.

However, this paper suffers from several limitations. Firstly, the algorithm violates the privacy principle of federated learning, since all the prototypes of each client are sent to all the clients. In addition, the t-test indicates that the difference between FedPCL and FedProto are not statistically significant for many cases in table 2.

**Award:**

No

---

### Decision · Program_Chairs · 2022-09-14

Accept